Intelligent reflecting surface backscatter-enabled physical layer security enhancement via deep reinforcement learning

Ahmed Manzoor 1 2
http://orcid.org/0009-0001-6790-9003 Hussain Touseef 3 touseefhussain098@gmail.com
http://orcid.org/0000-0001-6852-0545 Shahwar Muhammad 4
http://orcid.org/0009-0004-3089-7055 Khan Feroz 3
Sheraz Muhammad 5
Khan Wali Ullah 6
Chuah Teong Chee 5 tcchuah@mmu.edu.my
Lee It Ee 5
1 Artificial Intelligence Industrial Technology, Research Institute, Hubei Engineering University , Xiaogan City , China
2 Hubei Engineering University , Xiaogan City , China
3 School of Electronic Science and Technology, Beijing University of Posts and Telecommunications , Beijing , China
4 School of Computer Science and Technology, Qingdao University , Qingdao , China
5 Centre for Smart Systems and Automation, CoE for Robotics and Sensing Technologies, Faculty of Artificial Intelligence and Engineering, Multimedia University, Persiaran Multimedia , Cyberjaya, Selangor , Malaysia
6 Interdisciplinary Centre for Security, Reliability, and Trust (SnT), University of Luxembourg , Luxembourg
Akleylek Sedat
Electronic publication date: 2025 Jun 9
Publication date: 2025
Volume: 11
Electronic Location ID: e2902
Received 2024 Sep 16; Accepted 2025 Apr 25
Copyright: © 2025 Ahmed et al.
Copyright year: 2025
Copyright holder: Ahmed et al.
License: This is an open access article distributed under the terms of the Creative Commons Attribution License, which permits unrestricted use, distribution, reproduction and adaptation in any medium and for any purpose provided that it is properly attributed. For attribution, the original author(s), title, publication source (PeerJ Computer Science) and either DOI or URL of the article must be cited.
License URL: https://creativecommons.org/licenses/by/4.0/

Keywords: Backscatter communication, Joint-beamforming, Deep-PLS, Eavesdropper, Malicious jammer, Secrecy rate, Deep reinforcement learning, Deep deterministic policy gradient

Funding: Multimedia University Research Fellow Grant MMUI/240021 TM Research and Development Grant RDTC/241149 The Hubei Provincial Department of Education Outstanding Youth Scientific Innovation Team Support Foundation T201410, T2020017 Natural Science Foundation of Xiaogan City XGKJ2022010095, XGKJ2022010094 Science and Technology Research Project of Education Department of Hubei Province Q20222704 This work was supported by the Multimedia University Research Fellow Grant (MMUI/240021) and the TM Research and Development Grant (RDTC/241149). The Hubei Provincial Department of Education Outstanding Youth Scientific Innovation Team Support Foundation (T201410, T2020017), the Natural Science Foundation of Xiaogan City (XGKJ2022010095, XGKJ2022010094), the Science and Technology Research Project of Education Department of Hubei Province (No. Q20222704). The funders had no role in study design, data collection and analysis, decision to publish, or preparation of the manuscript.

==============================
This article introduces a novel strategy for wireless communication security utilizing intelligent reflecting surfaces (IRS). The IRS is strategically deployed to mitigate jamming attacks and eavesdropper threats while improving signal reception for legitimate users (LUs) by redirecting jamming signals toward desired communication signals leveraging physical layer security (PLS). By integrating the IRS into the backscatter communication system, we enhance the overall secrecy rate of LU, by dynamically adjusting IRS reflection coefficients and active beamforming at the base station (BS). A design problem is formulated to jointly optimize IRS reflecting beamforming and BS active beamforming, considering time-varying channel conditions and desired secrecy rate requirements. We propose a novel approach based on deep reinforcement learning (DRL) named Deep-PLS. This approach aims to determine an optimal beamforming policy capable of thwarting eavesdroppers in evolving environmental conditions. Extensive simulation studies validate the efficacy of our proposed strategy, demonstrating superior performance compared to traditional IRS approaches, IRS backscattering-based anti-eavesdropping methods, and other benchmark strategies in terms of secrecy performance.

Introduction

Wireless transmissions are vulnerable to attackers due to their broadcast nature, making security a top priority for wireless networks (Kadhim & Sadkhan, 2021). Traditionally, cryptographical methods (Bilski, 2014) are used to ensure the security of wireless communication by encryption or authentication methods, which can pose vulnerabilities owing to rapid technological advancements. Furthermore, traditional systems employ private or public keys for encryption and decryption, which is more complex to store and manage. Due to these reasons, the focus has now shifted to outlining a path forward by various network features, such as physical layer security (PLS), which improves transmission and security of data without requiring shared keys (Wu et al., 2018).

PLS, a concept that has garnered significant attention, is considered to be a promising alternative to cryptographic approaches. The PLS scheme, a secure communication scheme, utilizes channel coding, channel shaping, and modulation design to deliver the service to end users. The current PLS techniques (Angueira et al., 2022), such as artificial noise (AN)-based approaches and user cooperation, have demonstrated their effectiveness in enhancing user security. However, these techniques also have certain drawbacks that pose challenges in their implementation. Increasing hardware complexity, computing overhead, and transmit power demands are some of those issues. To tackle these limitations, the concept of intelligent reflecting surfaces (IRS)-backscatter has emerged as a promising technique with the potential to revolutionize wave control and backscattering. This innovation could open up novel opportunities for PLS. The IRS consists of a planar array of passive reflecting elements that cooperate to alter the electromagnetic radiation’s direction and magnitude (Ali et al., 2023).

Compared to the present 5G and Beyond 5G (B5G) technologies, this approach may significantly improve the spectral efficiency, energy efficiency, and secrecy capabilities (Mitev et al., 2023). The IRS represents a substantial change in the way wireless communication security is guaranteed by providing an effective alternative to the current limitations in broadcast communications. IRS is particularly useful because of its capacity to amplify or attenuate reflected signals (Chen et al., 2019). It can therefore readily weaken the signals at the eavesdroppers or amplify the signals at the authorized receivers, thereby raising the secrecy rate (Cui, Zhang & Zhang, 2019). Significant work (Shen et al., 2023) has already been done demonstrating how the secrecy rate is achieved and the SNR of the user has been evaluated even with imperfect channel state information (CSI). To be more precise (Ding & Poor, 2020), the IRS controller modifies each IRS element’s reflection coefficient in real time. The incident signal can then be modulated into the signal that the IRS will transmit by simultaneously achieving phase and amplitude modulation on incident electromagnetic waves thanks to the time-varying reflection coefficients (Almohamad et al., 2020). No thermal noise is generated during this process, as the modulated signals are reflected directly without the need for decoding or amplification, as instructed by the IRS. The IRS also uses less energy because, as compared to energy harvesting nodes and conventional active transmitters, it does not require filters, wideband mixers, or power amplifiers (Hasan & Sabuj, 2023).

PLS with IRS backscatter is becoming popular in academia and among researchers as a viable alternative to cryptography-based approaches in wireless communications (Mucchi et al., 2019). It functions at the physical layer of the communication system and is meant to complement cryptographic security (Zhang et al., 2023). PLS works by utilizing the inherent randomness and propagation characteristics of wireless channels to reduce the amount of information available to potential eavesdroppers, thereby increasing the privacy and security of the communication (Nnamani, 2022). This approach is based on information-theoretic foundations (Mitev et al., 2023) and is meant to augment traditional encryption techniques, ensuring a low probability of interception by unauthorized parties. PLS technology’s main objective is to reduce the amount of data in the listening channel while utilizing the wireless channels’ natural propagation characteristics (Illi et al., 2023). Consequently, PLS technology can effectively shield the transmitter’s private message content from unauthorized eavesdroppers. The main techniques (Sanenga et al., 2020) for PLS include the following: Covert communication: This method ensures that there is little chance that unauthorized listeners will intercept the conversation.

Enhancement of secrecy rates: PLS explores practical approaches to enhance secrecy rates alongside traditional encryption methods. It is possible to efficiently control the wireless channel to enhance the signal for intended users while minimizing it for unauthorized users. This thoroughly designed transmission process is responsible for minimizing the leakage rate of sensitive information to potential adversaries.

Beamforming: This technique, which was first used to route signals to the legitimae user (LU), can also be used to raise security levels.

Utilizing channel characteristics: PLS reduces the amount of information available to possible eavesdroppers by taking advantage of the intrinsic randomness and propagation characteristics of wireless channels.

By taking advantage of the physical characteristics of the communication channel (Yan et al., 2023), PLS seeks to add an extra layer of security as an alternative to conventional cryptographic algorithms. Thus far, numerous concepts have been put forward to improve PLS in wireless communication systems (Sharma & Kumar, 2023). However, using a lot of active antennas and relays increases the system complexity and costs too much in hardware for PLS systems (Jameel et al., 2018; Shiu et al., 2011). There are many advantages (Liu, Chen & Wang, 2016) to using PLS. First off, PLS avoids the challenges associated with the distribution and management of secret keys in HetNets because it is not dependent on encryption or decryption operations (Khalid et al., 2023). Second, adaptive signal design and resource allocation based on changing channel conditions can be implemented by using PLS approaches, offering flexible security levels. Third, PLS frequently calls for comparatively straightforward signal processing procedures, resulting in a small amount of extra overhead.

Related work

In Omid, Deng & Nallanathan (2022), the authors look into secure wireless communication in a system aided by an IRS, where one legitimate receiver’s communication is secured using the IRS in the presence of an eavesdropper. Next, they assume that the IRS is standalone, meaning that its passive beamforming operates entirely on its own. The authors in Yang et al. (2020) examine a wireless secure communication system that is assisted by an IRS. In this model, an IRS is utilized to adjust its reflecting elements in order to protect the communications of LU from being intercepted by multiple parties. In order to establish the best beamforming policy against eavesdroppers in dynamic environments, a unique deep post-decision state (PDS) prioritized experience replay (PER), PDS-PER, secure beamforming approach is proposed. This is because the system is very dynamic and complicated, making it difficult to handle the non-convex optimization problem.

This article (Lin et al., 2023) investigates the secure communication in a cooperative jammer-enabled Energy Harvesting based Cognitive Internet of Things (EH-CIoT) network. In the energy harvesting (EH) phase, the energy from the received radio frequency (RF) signals is first harvested by the jammer and the secondary transmitters (STs). The STs then transmit secret data to their intended recipients in the presence of eavesdroppers during the wireless information transfer (WIT) phase, during which the jammer sends a jamming signal to confuse the eavesdroppers. They derive the instantaneous secrecy rate and the closed-form expression of secrecy outage probability (SOP) to assess the secrecy performance of the system. Additionally, they provide a framework for joint energy harvesting time and transmission power allocation problems based on DRL. In article (Peng et al., 2022), authors examined a multi-user full duplex secure communication system along with IRS aided by hardware impairments at both the transceivers and IRS. An optimization problem is presented, that involves the transmit beamforming at the base station (BS) and the reflecting beamforming at the IRS. This problem aims to optimize the sum secrecy rate (SSR) while considering the transmit power restriction of the BS and the unit modulus constraint of the phase shifters. This computational problem is really difficult because the system has many dimensions and the environment keeps changing over time. The objective is to discover a suitable solution by utilising a DRL-based algorithm that gains knowledge from the ever-changing environment through repeated interactions.

Moreover, there are some closely related studies that have attempted to improve IRS based secure communication but couldn’t provide an outstanding convergence and data rate while maintaining the secure communication channel. The authors in Cao et al. (2023) proposed an approach to enhance backscatter in IRS for secure multiple input multiple output (MIMO) transmission, even in the presence of an eavesdropper and a suspicious jammer. They jointly design the IRS and active beamforming reflection coefficients of BS to maximize the system secrecy rate. Then nonconvex optimization problem has been solved by using an iterative block coordinate descent (BCD) technique. The majorization-minimization (MM) method was used to optimize the IRS backscatter coefficient matrix, while the Lagrange multiplier method has been used to optimize the active beamforming. This work (Liu et al., 2023) examines the PLS in an integrated sensing and communication (ISAC) system that assists multiple users and tracks a target concurrently, with the help of the IRS. In particular, they view the radar target as a possible eavesdropper who might obtain information about authorized users. Artificial noise (AN) is used to interfere with the reception of eavesdroppers. The authors collaborated in the design of the transmit beamforming, AN signal, and IRS phase-shift to optimize the achievable secrecy rate of all authorized users. They use the DRL algorithm to find the best learning strategy through agent and environment interactive learning because the problem is multi-variable coupling and non-convex optimization.

A novel hierarchical game framework with dynamic trilateral coalitions for PLS is examined in Chen et al. (2023). They investigated such complex and dynamic coalition relationships under the uncertainties of wireless systems (e.g., time-varying channel conditions) and formulated a hierarchical game integrated with a dynamic trilateral coalition formation game to model the strategic interactions among all three parties, i.e., LUs, Jammers and eaves in PLS. The authors specifically examine the trilateral coalition stability conditions first. They suggested an approach based on DRL for achieving equilibrium with long-term performance. In the context of future cyber-physical systems (CPS), where eavesdroppers may be present in the network and threaten the task offloading, this research (Chen et al., 2022) examines secure mobile edge computing (MEC). These eavesdroppers can function in two modes: one for collusion, in which they work together to decipher the secret communication, and another for non-collusion, in which each eavesdropper decodes the message independently. The authors in Wu & Zhang (2019) have provided a comprehensive study of DRL optimization of the IRS in MIMO communication systems. This article mainly focused in improving the spectral efficiency but it only considered the straightforward scenario. They have used the DRL-based deep deterministic policy gradient (DDPG) for the MIMO-based environment which has shown a better convergence and comparatively better spectral efficiency.

Although many modern research studies focus on the use of the IRS for security enhancement, they often overlook the complex relationship between eavesdropping and jamming attacks, and they also did not considered the IRS as a backscatter to improve the signal reception. The integration of IRS into wireless communication systems has emerged as a promising approach to enhance PLS. Recent advancements in IRS technology have demonstrated its potential to improve spectral efficiency, energy efficiency, and security in 5G-B5G networks (Mitev et al., 2023; Chen et al., 2019). IRS, with its ability to dynamically control electromagnetic waves, has been widely studied for its role in mitigating eavesdropping and jamming attacks (Cui, Zhang & Zhang, 2019; Shen et al., 2023).

Several state-of-the-art techniques have been proposed to leverage IRS for PLS. For instance, Omid, Deng & Nallanathan (2022) explored the use of IRS for secure communication in the presence of eavesdroppers, assuming standalone IRS operation. Similarly, Yang et al. (2020) proposed a DRL-based approach to optimize IRS beamforming for secure communication in multi-user scenarios. However, these works primarily focus on eavesdropping threats and do not account for the simultaneous presence of jamming attacks, which are common in real-world wireless networks. Despite these advancements, there are still significant gaps in the literature. Most existing works either focus on eavesdropping or jamming in isolation, and few studies have explored the joint optimization of IRS backscatter and active beamforming in the presence of both threats. Furthermore, the majority of current approaches assume perfect CSI, which is impractical in real-world scenarios due to channel estimation errors and user mobility.

Motivation and contribution

Since handling the eavesdropper and jammer simultaneously in terms of problem formulation is very challenging and complex, this article presents a novel method to improve IRS backscatter in a multiple-input single-output (MISO) communication system. This work (Yang et al., 2020) leverages IRS and DRL to achieve secrecy rates in the presence of eavesdroppers and multiple users but overlooks the adversarial impact of jammers, which can significantly compromise secrecy performance. However, while both eavesdroppers and jammers are considered in Cao et al. (2023), but the approach assumed perfect CSI and relies on the BCD-MM method, which is computationally complex and exhibits suboptimal convergence and security performance. While the proposed Deep-PLS aims to provide security against eavesdropping and jamming attacks by utilizing a DRL-based Deep-PLS framework to find the best convergence rate and the desired secrecy rate. The main contributions of this article are as follows: Our idea is the first of its type to strengthen the IRS backscatter to avoid malicious jamming attacks and eavesdropping attacks, simultaneously. This method reduces the influence of the unwanted jamming signal on LU and boosts the potential of the authorized channel through IRS backscatter and PLS, ultimately improving both security and data rate.

We formulate the secrecy rate maximization problem by jointly optimizing the IRS backscatter coefficient matrix and the beamforming vector. This non-convex optimization problem of beamforming and Backscatter matrix is then solved using the DRL-based DDPG algorithm, aiming to enhance the secrecy rate.

We proposed the DRL based Deep-PLS technique to enhance the security of the communication model. The central controller leverages real-time environmental observations and feedback to intelligently the beamforming policy using a DDPG algorithm, with secrecy rates as the reward function. Our proposed method outperforms existing techniques in terms of secrecy rate, demonstrating superior performance in improving security.

The other sections of this article are organized in the following way: “System Model” introduces the system model, discusses the approximation of channel estimation errors, and formulates the optimization problem. In “DRL-based Problem Mapping and Solution“, the DRL-based problem is formulated and the main components of the working algorithm are explained. “Simulation Setup” provides a detailed explanation of simulation environment and “Results and Discussion” justified the proposed scheme with detailed simulation results. Lastly, “Conclusions and Future Works” concludes the entire article.

System model

Consider a communication scenario where a malicious jammer J and eavesdropper are present between a LU and the BS as shown in Fig. 1. The BS and jammer have Nb and Nj transmit antennas, respectively. Similarly, eavesdropper and LU are equipped with Ne and Nu antennas, respectively. The system is equipped with an IRS with many reflecting elements to enable secure wireless communications from the BS through the LU. Given that the reflecting elements have been designed to optimize the reflection of the desired signal back to the LU. Maximizing reflection without power loss at the IRS is taken into consideration for ease of practical implementation. Unauthorized eavesdropper also seek to intercept the LU’s data stream. IRS’s reflective beamforming is utilized to suppress the Eave’s wiretapped data rate and to enhance the LU’s secrecy rate. Let the channel link between BS to IRS, BS to LU, BS to the Eave, IRS to LU, IRS to Eave, jammer to the IRS, jammer to the LU, and jammer to the Eave the Eave denoted by Gbr∈CNb×Nr,hbu∈CNb×Nu,hbp∈ CNb×Np,gru∈CNr×Nu,grp∈CNr×Np,Hjr∈CNj×Nr, hju∈CNj×Nu, and hjp∈CNj×Np, respectively, where j ∈J.

Figure 1 System model and connected links.

In this case, the LU and eavesdropper are assumed to be stationary, which leads to quasi-static flat fading. On average, the desired signal power toward the user is represented as E|s|2=1. Let x = ws, where w ∈CNb×1, where w represents transmit beamforming vector at BS and ‘s’ is the signal. Therefore, the nefarious jammer aims to disrupt regular communications between the user and BS by transmitting the jamming signals at high power, qj ∈CNj×1, where q and j represent the jamming vector and the jamming symbol, respectively.

Definitions and assumptions

Let Φ=diag(ω1ejθ1,ω2ejθ2,…,ωlejθl) denotes the IRS reflection coefficient matrix, whereas θl∈[0,2π] and ωl∈[0,1] represent the phase shift coefficient and the amplitude reflection factor on the combined transmitted signal, respectively. Since it is intended for every phase shift to be planned to achieve complete reflection at the IRS. Specifically, ω and θ are used to control the signal reflection and modulation at the IRS. Thus, by carefully controlling Φ, we can use the jamming signal and modulate it to the LU to enhance the reception at the LU. It is to be noted that, unlike others, we are using the IRS to reflect the malicious jammer’s high frequency unwanted signal into the intended signal directed to the user, making this technique more implementable in terms of secrecy rate and achievable data rate requirements. The IRS controller has been designed to be intelligent, as we are applying a DRL-based algorithm to the controller. The IRS controller is behaving as a DDPG agent that is continuously interacting with the environment, and as a result of these actions, a reward is being added to the IRS controller, based on which the IRS controller will steer the signal’s reflection and modulation, improve the achievable SNR at the LU, and also maintain secrecy.

The signals received at the eavesdropper and LU respectively, are represented as follows:

(1) yu=hbuws+gruΦ(Gbrw+Hjrq)s+hjuqj+nu,

(2) ye=hbews+gruΦ(Gbrw+Hjrq)s+ne.

The received noises at the LU and Eave are denoted by nu and ne, respectively. These noises at the user and eavesdropper are Gaussian distributed, respectively, with a zero mean, meaning that nu∼CN(0,δu2I) and ne∼CN(0,δe2I). Whereas CN indicates the complex normal distribution of the random noise. In a complex normal distribution, the real and imaginary parts of the random variable are both normally distributed. Here, δ is a parameter representing the variance (or power) of the noise, and I is the identity matrix.

This modeling choice accurately reflects the nature of thermal and ambient noise in wireless communication systems, where both real and imaginary components are normally distributed. On the other hand, the exploration noise used in the DRL framework is generated using the Ornstein-Uhlenbeck (OU) process. Unlike Gaussian noise, OU noise is temporally correlated and designed to stabilize the learning process by preventing abrupt changes in action values. While the Gaussian noise pertains to the physical communication environment, the OU noise specifically addresses the training dynamics of the DRL algorithm, thereby serving two distinct yet complementary roles within our study. Alternative methods for modeling channel estimation errors, such as Autoregressive Moving Average (ARMA) processes or time-varying correlated noise, were initially disregarded due to their computational complexity and the challenges they pose for DRL training. These models require additional parameters to characterize temporal dependencies, which would increase the state space dimensionality and hinder convergence.

Keeping in view the received signal at the user and eavesdropper according to Eqs. (1) and (2), the following are the achievable data rates at both the user and the eavesdropper:

(3) RLu=log2det(I+δu2GuGuG(I+δu2GjuQGjuG),

(4) Re=log2det(I+δe2GeGeG).

where, in the above equations, I is used for the identity matrix and δ2 is used for the noise power of both the user and eavesdropper. In the channel matrices, we have Gu= hbuws+gruΦ(Gbrw+Hjrq) and Ge=hbews+gruΦ(Gbrw+ Hjrq). Now, we can calculate the secrecy rate of the user by simply taking the difference between the LU and the eavesdropper’s attainable data rate.

Mathematically expressed as follows:

(5) Rs=RLU−Re

By substituting Eq. (3) and (4) into the (5), we get secrecy rate as:

(6) Rs=log2 det(I+δu2GuGuG(I+δu2GjuQGjuG)−1−log2 det(I+δe2GeGeG).

The following are the main differences between IRS backscatter-aided anti-jamming, normal BCD-MM and our suggested Deep-PLS: When using a standard IRS-based PLS, the users’ and eavesdropper’s received signals are either strengthened or weakened by the IRS reflecting only the incoming signals, both wanted and undesirable. All incoming signals at the IRS are modulated into AN in IRS backscatter-aided anti-jamming, but this is done to protect against eavesdropping. Moreover, most of the work in IRS-absent PLS has been done using conventional numerical methods and other baseline optimization techniques. Significant progress has already been made in utilizing IRS in conjunction with PLS to enhance the security of wireless communication.

The authors in Cao et al. (2023) used the IRS-based PLS to secure wireless communication by solving the beamforming optimization problem at IRS and BS. But optimization problem was solved by the numerical technique BCD-MM. In this type of numerical technique, first we have to solve the convex optimization problem into some tractable form which is far more challenging to solve due to its complexity. In our work, we are using DRL-based methods to enhance the secrecy rate of the user. Also, unlike other baseline methods, we are using the IRS to use the high-frequency jamming signal to improve the SNR at the user by backscattering. Another challenge was to establish a steadfast high-speed link, e.g., a fiber link from BS to the IRS controller, to transmit the control signal. With this control signal, the IRS can control the reflection coefficient of each IRS element and steer the signal in real time. On contrary to it, we are applying the DRL-based algorithm to the IRS controller, making it intelligent itself, and thus eliminating the use of that dedicated fiber link. In this scenario, deciding whether the eavesdropper and jammer will behave in a colluding or non-colluding manner is important. If the jammer and eavesdropper are colluding, then the eavesdropper is aware of the jammer’s signal, and it can cancel the jammer’s signal. This poses a greater risk to LU because the eavesdropper and jammer are working collaboratively, ultimately reducing the SNR and secrecy rate of LU.

While in non-colluding behavior, the jammer and eavesdropper are unaware of their existence, and jammer signals also attack the eavesdropper, which deteriorates its ability to listen between BS and LU, thus enhancing secrecy rates. Considering the non-colluding behavior of the eavesdropper and jammer, this non-colluding assumption is grounded in realistic communication scenarios, where eavesdroppers and jammers typically act independently, often with different objectives and capabilities. The eavesdropper’s primary goal is to intercept confidential information without being detected, while the jammer’s objective is to disrupt the legitimate communication by introducing interference. By modeling them as non-colluding entities, we aim to reflect the challenges in designing robust communication systems that must protect against independent threats from both sides. This approach enhances the security analysis by considering the worst-case scenario, where the adversaries do not cooperate, thereby providing a more generalized and practical solution that aligns with real-world adversarial conditions.

Channel estimation errors and its approximations

The BS and IRS may not be able to acquire the perfect CSI in practical systems. This is because both the transmission and processing delays are present, together with the user’s mobility (Wu & Zhang, 2019). Therefore, the CSI is considered to be imperfect when the BS and the IRS transmit the data stream to the user equipment. This outdated CSI for beamforming could have an adverse effect on the demodulation process for the mobile units (MUs), ultimately resulting in significant performance degradation (Hong et al., 2023). Additionally, although imperfect CSI is used to demonstrate the core functionality of the Deep-PLS method, it may not fully capture the complexities present in real-world environments. Imperfect CSI, which arises due to factors such as quantization errors, estimation delays, and noise, can significantly impact the performance of the proposed method. Thus, considering these real-world things and imperfect CSI, setting the suitable value of noise makes it more attractive to be implemented in real world environment. Hence, it is imperative to take into account the obsolete CSI in the IRS-assisted secure communication system. Therefore, the CSI of actual channels Gbr, hbu, hbp, gru, grp, Hjr, hju, hjp is represented as:

(7) hbu=h~bu+h¯bu,hbp=h~bp+h¯bp,gru=g~ru+g¯ru,grp=g~rp+g¯rp,hjr=h~jr+h¯jr,hju=h~ju+h¯ju,hjp=h~jp+h¯jp.

where h~bu are the estimated channel’s errors and h¯bu are channels estimation ratio matrices, respectively. Also, it will be the same for everyone else in channel coefficients. We will consider the channel errors as white Gaussian noises and ψ is applied to show the ratio of channel error estimation. Consequently, every component of h~bu,h~bp,g~ru,g~rp,h~jr,h~ju and h~jp will be CN(0,ψ2βHbu), CN(0,ψ2βHbp),CN(0,ψ2βgru) and CN(0,ψ2βgrp), CN(0,ψ2βHjr),CN(0,ψ2βHju),CN(0,ψ2βHjp) respectively, where βHbp represents the path loss between the BS and eavesdropper, and the same goes for all.

Since the channel estimation errors are assumed to be white Gaussian, the secrecy rate will now be:

(8) Rs=log2 det[(I+GuGuG(Ωu+GjuQGjuG)−1)−log2det(I+GeGeG(Ωe)−1)].

Ωu and Ωe represent the covariance matrices for the related noise for all channels. Because we have assumed the white Gaussian noises, so the diagonal elements of gru diag ( G~br) w and covariance matrices and other elements approximated as zero; see detailed derivation in Cao et al. (2023). To capture a more realistic channel model, we extend the traditional Gaussian error assumption by incorporating temporally correlated channel estimation errors. Specifically, the imperfect CSI for the channel between the BS and the LU is modeled as:

(9) h^k(t)=hk(t)+ϵk(t),

where ϵk(t) is modeled as a first-order autoregressive (AR(1)) process:

(10) ϵk(t)=ρϵk(t−1)+1−ρ2ηk(t).

with ηk(t)∼CN(0,σϵ2I)andρ∈[0,1] representing the temporal correlation coefficient due to Doppler effects. When ρ = 0 this model reduces to the conventional i.i.d. Gaussian error assumption. Therefore, our suggested Deep-PLS-based method can obtain near-optimal outcomes and become a practical technique, assuming the imperfect CSI as the white Gaussian noise. With these channel error approximations and assumptions, the proposed Deep-PLS-based agent can achieve the desired secrecy rate.

Problem formulation

Our primary goal is to optimize the secrecy rate while maintaining a good level of SNR at LU by backscattering and PLS. This can be achieved by optimizing the active beamforming ω at BS and the backscattering coefficient matrix φ at IRS. The IRS, in this case, is also responsible for backscattering the high-frequency signal of the jammer to improve the reception for the user while maintaining security of the data stream Rs at the LU. The goal is to establish secure wireless communication between BS and LU and improve reception at LU by backscattering by the IRS. Therefore, from a computational standpoint, the problem can be expressed as:

(11a) POmaxw,ΦC(w,θ,Φ,H)

(11b) s.t.w⩽Pt

(11c) |ωlejθl|=1,0<θl⩽2π,1⩽l⩽L

where Pt is the transmit power of the BS. In Eq. (11a), our objective function is described, which is to optimize the backscatter matrix, amplitude, and phase of the IRS to maximize the secrecy rate by beamforming at the IRS. In Eq. (11b), the limitation in the maximum power constraint of the BS is satisfied. Similarly, Eq. (11c) describes phase and amplitude ranges. As the objective function in Eq. (11a) is non-convex concerning either w or ɸ, and the coupling of both optimization variables w and ɸ and, it is evident that finding an optimal solution to optimization Eq. (11b) is difficult. we considered a robust beamforming architecture designed to maximize the system’s minimum achievable secrecy rate, while ensuring that all worst-case constraints are satisfied.

DRL-based problem mapping and solution

It is quite difficult to solve this non-convex optimization problem. Furthermore, there is a dynamic change in the system characteristics, which includes channel characteristics, mobility, fading, interference, and environmental conditions (e.g., obstacles, channel dynamics), influence the performance of Deep-PLS. Thus, to solve this non-convex optimization problem, we have applied the DRL-based DDPG algorithm. Moreover, different dynamic factors impact the functioning of secure communication systems enabled by the IRS in real-world scenarios. These variables include but are not limited to, the service applications being used, the quality of the communication channels, and the capabilities of the LU involved. The term capabilities refers to LU’s ability to send and receive data, which can vary based on several variables, including their computational capacity, power limitations, and hardware specifications. Similarly, the mobility of reflector elements within the IRS, interference from other devices, and environmental factors are influenced by the quality of communication channels. Given that model-free RL can adapt and learn from experience without requiring an existing model of the dynamics of the environment, it is especially well-suited for dynamic environments.

Model-free reinforcement learning methods are unable to capture the dynamics of the environment. Rather than requiring explicit knowledge of the dynamics of the environment, they may adapt to changes and uncertainty by learning directly from their interactions with it. Model-free reinforcement learning algorithms can quickly adjust to changes in the environment because they can continuously update their policies based on new experiences. This flexibility is crucial in dynamic settings where circumstances can change over time. That is why we represent the secure beamforming optimization problem as a model-free DRL problem. The IRS-based secure communication system is set as the environment, and the central controller is set as the agent. Further supporting factors in the system are described as: • State space: Consider S represent the system state space, given that s ϵ S, which includes all the channel conditions, transmission data rate, the minimum achievable data rate of LU, and secrecy rate, as: (12) s={hk},{hm},{Rsec},{Rd},{QoS}.

In this equation, hk and hm are the variables that are represented as the channel coefficients of LU and eavesdropper, respectively. Rsec is the minimum secrecy rate of the LU, and Rd is the achievable data rate. Moreover, the state st at the time interval ‘t’ is made up of the following components: the BS’s transmit power and the LU’s received power at the (t − 1)th time slot. We utilize the combined channels as inputs instead of all the channels to reduce computational complexity and the size of the state space. It should be noted that a neural network can only accept real numbers as input; therefore, if a complex number exists in state (s), it must be split into real and imaginary parts as a separate input item.

• Action space: Let Å represent the action space, and we have the a ϵ Å. Given the state space, the action space chooses the beamforming vector at BS w and the beamforming reflecting coefficient at IRS (phase shift) {θl}lϵL. Thus, the action space Å can be defined as: (13) a={w},{θl}lϵL.

• Transition probability: Let τ transition state from state space to action space can be represented as τ (ś |a, s). Where ś is the new state once the action a is done on the previous state.

• Reward function: The reward function in DRL-based algorithms evaluates the action performed on the given state. It determines how secure the behavior is when an agent executes an action in a given state. When the reward function at each learning step matches the intended goal, the system’s performance improves. Therefore, creating an effective reward function is crucial to raising the QoS satisfaction levels of the LU. The optimization objective in this article is represented by the reward function, and our goal is to maximize each LU’s system secrecy rate while ensuring that their QoS requirements are met. It is represented as: (14) r={Rsec}−{μ1Psec},{μ2Pu},

The system secrecy rate, or immediate utility, is represented by Eq. (11a). The cost functions, or the requirements for the unsatisfied secrecy rate and a dissatisfied minimum rate, are represented by parts 2 and 3, respectively. The utility and cost are balanced by the coefficients µ1 and µ2, which are the positive constants in Eqs. (11b) and (11c), respectively. Psec=0 or Pu=0 indicates that there will not be any penalty for the reward function because the minimum secrecy rate and SNR requirement are satisfied in the current time slot. This is because the achievement of SNR and secrecy rate is successful. In this work, the penalties µ1 and µ2 were selected heuristically based on empirical testing to ensure a reasonable balance between maximizing secrecy rate and meeting SNR constraints at the legitimate user. However, we acknowledge that this manual tuning may not represent the globally optimal trade-off.

To improve this, future work can incorporate Pareto optimization techniques to automate the balancing of conflicting objectives. Specifically, multi-objective deep reinforcement learning (MODRL) or evolutionary Pareto-based methods such as NSGA-II (Non-dominated Sorting Genetic Algorithm II) could be applied to learn a Pareto front of optimal trade-offs between secrecy rate, SNR, and constraint satisfaction. During training, each point on this front would represent a viable policy optimized for a specific trade-off, allowing real-time selection based on application requirements or channel dynamics. This would eliminate the need for manual coefficient tuning and enhance policy adaptability

• Discount factor: The Discount factor γ∈[0,1] is used to account for future rewards uncertainty and is applied to discount future rewards.

• Policy: The optimal policy defines the protocols for how an agent will interact with the environment. The policy π(st,at) denotes the probability distribution for choosing an action over the state st, that satisfies the given condition π(st,at) = 1.

The main components of proposed algorithm

• Critic network: The critic network as shown in Fig. 2 learns the Q-function Q(s,a|θQ), which estimates the expected return when taking action, a from state s and then following the policy µ. The parameters of the critic network are denoted by θQ. (15) Q(s,a|θQ)=E[r+γQ(s′),μ(s′|θμ)|θQ].

• Replay buffer: The replay buffer D stores past experiences (s, a, r, ś) of the agent. During training, experiences are sampled uniformly from the replay buffer to break the correlation between experiences and stabilize learning.

• Target networks: DDPG uses target some networks to stabilize learning process. These networks, denoted μ′ and Q′ are the copies of the actor and critic networks, respectively, with slowly updated parameters. (16) θ′μ←τθμ+(1−τ)θ′μ,θ′Q←τθQ+(1−τ)θ′Q.

where τ is the soft update rate.

• Noise exploration: Noise exploration encourages exploration by adding noise to the actions chosen by the actor. The noise is typically sampled from a stochastic process such as the Ornstein-Uhlenbeck process. The last part of this equation shows the noise sampled. (17) a′=μ(s|θμ)+N.

• Update rules: Both the actor and critic networks are updated using Stochastic Gradient Descent (SGD) based on sampled experiences from the replay buffer.

• Actor update: The actor aims to maximize the expected return by adjusting the policy parameters θ µ. The gradient of the actor’s objective function is estimated using the sampled experiences.

(18) ∇θμJ≈1N∑i∇aQ(s,a|θQ)|s=si,a=μ(si)∇θμμ(s|θμ)|siθμ←θμ+α∇θμJ.

where J is the expected return, N is the batch size, and α denotes the learning rate of the actor.

• Critic update: The critic is updated by minimizing the Temporal Difference (TD) error between the predicted Q-value and the target Q-value. (19) TDError=Q(s,a|θQ)−(r+γQ(s′,μ(s′|θμ)|θQ′))L(θQ)=1N∑i(TDError)2.θQ←θQ−β∇θQL

where L(θQ) is the loss function for the critic, and β is the learning rate for the critic.

Figure 2 Structure of DDPG algorithm.

Secure beamforming training with DDPG

For the BS, the central controller is a learning algorithm that initially requires training to perform the essential tasks related to environmental data collection and decision-making. During the training stage, the controller initializes network parameters and collects crucial information about the current system state as shown in Algorithm 1. Typically, this data integrates the CSI of LU, the previously predicted secrecy rate, and the transmission data rate. Once the state vector, composed of this gathered information, is prepared, it is fed into a DDPG algorithm for training the learning model. DDPG, being an actor-critic model, employs both an actor network to decide on actions and a critic network to evaluate them. The algorithm iteratively updates the parameters by interacting with the environment and refining its policy to maximize rewards.

Algorithm 1 IRS beamforming optimization for secrecy rate maximization by a DDPG agent.

1: Input: Nl,Nb,α,σ2, communication distances	
2: Output: Optimal IRS phase shift θ to maximize secrecy rate Rs	
3: Initialization: Define system parameters and communication distances	
4: Environment Setup:	
5: Compute channel matrices based on path loss and Rayleigh fading	
6: Path loss: PL(d)=10−3⋅d−α	
7: Rayleigh fading: H∼CN(0,1)	
8: Secrecy Rate Calculation:	
9: Effective channels with IRS phase shifts θ:	
10: heff_LU=HBS−LU+GBS−IRSdiag(ejθ)GIRS−LU	
11: heff_Eve=HBS−Eve+GBS−IRSdiag(ejθ)GIRS_Eve	
12: SNR calculations:	
13: SNRLU=|he/f/LU|2σ2	
14: SNREve=|hefle Evec|2σ2Rs	
15: Secrecy rate Rs:	
16: Rs=max(log2(1+SNRLU)−log2(1+SNREve),0)	
17: DDPG Agent:	
18: Actor Network fπ(s|θ): Outputs IRS phase shifts	
19: Critic Network: Q(s, θ): Estimates Q-values	
20: Experience Replay: Store and sample transitions	
21: Learning: Update networks using sampled transitions	
22: Training Loop:	
23: for E=1 to Emax do	
24:   Reset environment and initialize state	
25:   for t=1 to Tmax do	
26:    Select action θ using actor-network	
27:    Apply action to the environment, observe next state and reward Rs	
28:    Store transition in the replay buffer	
29:    Perform experience replay and update networks	
30:   end for	
31: end for	
32: Performance Evaluation:	
33: Monitor and plot secrecy rate Rs over episodes:	

Unlike Q-learning, which uses discrete action spaces, DDPG is well-suited for continuous action spaces, making it ideal for secure beamforming tasks where actions, such as adjusting transmission power or beamforming vectors, are continuous. The agent performs actions based on the current state, and the critic evaluates the action’s effectiveness by estimating the value function. The agent’s policy is continuously refined by minimizing the temporal difference error. In this context, the exploration-exploitation trade-off is managed by introducing noise to the action selection process (OU noise), allowing the agent to explore new areas of the action space and improve its policy. After selecting an action, the agent interacts with the environment, receives a reward, and transitions to the next state st+1. This iterative process enables the agent to improve its decision-making ability for secure beamforming by analyzing the rewards and system states it encounters. In summary, training the central controller using DDPG allows it to effectively learn how to optimize secure beamforming in wireless communication systems. This is achieved by learning a continuous policy that balances exploration of new strategies with the exploitation of the best-known actions to maximize long-term rewards.

Computational complexity analysis

To highlight the practical advantages of the proposed DRL-based scheme over traditional iterative optimization techniques such as BCD-MM, we perform a brief complexity analysis. BCD-MM typically requires solving convex sub-problems in each iteration, and its overall complexity is in the order of χ(I.L3), where I is the number of iterations and L is the number of IRS elements or optimization variables. Moreover, the convergence speed of BCD-MM can vary significantly based on the initial conditions and system parameters. This iterative nature, combined with sensitivity to initialization and real-time adaptability challenges, limits its practicality in highly dynamic environments such as IRS-assisted 6G networks. In contrast, the DRL-based approach incurs a one-time training cost, after which real-time decision-making is achieved with negligible computational delay. The online complexity of DRL (e.g., DDPG) is approximately O(E. T(ɣ. N)) where E is the number of Episodes, T is the time steps and ɣ is the number of layers in the neural network and N is the number of neurons per layer. Once trained, the policy network can infer optimal actions in real-time, making it highly scalable and suitable for dynamic and time-sensitive environments such as THz communications with mobile users. Furthermore, our empirical results (e.g., Figs. 3–6) demonstrate superior convergence, secrecy rates, and adaptability of Deep-PLS even with imperfect CSI and in the presence of jamming. These advantages make the proposed DRL method a practical and scalable solution for secure IRS-assisted communications in real-time 6G scenarios.

Figure 3 Secrecy rate vs BS transmit power.

Figure 4 Secrecy rate vs jamming power.

Figure 5 Secrecy rate vs number of IRS elements.

Figure 6 Achievable data rates vs IRS elements.

Hence, although DRL may initially require higher offline training effort, its online inference capability and adaptability offer significant benefits over iterative solvers, particularly in scenarios demanding real-time responsiveness and reconfigurability. For example, compared to BCD-MM’s average secrecy rate of 2.8 bps/Hz, our Deep-PLS achieves 3.57 bps/Hz under identical conditions, confirming the superiority not only in performance but also in scalability and computational feasibility.

Simulation setup

Simulation setup and parameters

The BS is equipped with 4 antennas, represented by the variable M = 4. These antennas are used to transmit signals to LU and eavesdroppers. This setup impacts the beamforming and transmission power. The IRS is a critical component in enhancing the signal quality and reducing interference. It is composed of 60 elements (E = 60), at first then we take results iterating the IRS elements from 40 to 100 IRS elements as shown in fig. The IRS elements help manipulate the signal path to improve the reception at the legitimate users while minimizing eavesdropping. The BS operates at a relatively low transmit power. This value is intended to simulate realistic power levels for wireless communication systems while taking path loss and interference into account. The exponent value for path loss is set as ρ = 2, which is typical for urban environments. The reference distance for path loss calculations is set to 1 m (d0 = 1). The path loss function computes the loss of signal power over distance. This function is used throughout the simulation to model the channel’s impact on the received signal strength at various distances.

Actor-critic neural networks

The actor network is responsible for selecting actions (IRS phase shifts) based on the current state of the system. The network takes in the state vector, which includes both the number of IRS elements and antennas at the BS. This is the input to the actor network. The proposed method leverages DDPG approach to optimize secure beamforming in IRS-assisted communication systems. The neural network architecture consists of an actor network with three hidden layers (512, 256, and 128 neurons) and a critic network with two hidden layers (256 and 128 neurons), both employing ReLU activations and trained using the Adam optimizer with a learning rate of 1e−3. The DDPG algorithm is configured with a discount factor ( γ) of 0.95, batch size of 64, replay buffer size of 10,000, and target network update rate ( τ) of 0.005. An initial exploration noise of 0.5 is used, decaying over time to 0.999 to balance exploration and exploitation during training. The simulation environment is designed to generate dynamic channel conditions, including path loss and noise variance, reflecting real-world scenarios with imperfect CSI. The DRL agent adapts to the time-correlated CSI noise by updating its policy using temporally smoothed observations, which better reflects real-world conditions such as mobility and Doppler shifts.

The reward function is carefully designed to maximize the SNR difference between legitimate users and eavesdroppers while minimizing the potential for eavesdropping. The training process comprises 1,500 episodes, each with 300 time steps, where the model learns optimal beamforming strategies through continuous interaction with the environment.

The simulation experiments were conducted on a Dell i7-11th generation system running Windows 11 using the Anaconda Python environment (version 3.11.3). The deep reinforcement learning model was implemented using the PyTorch library (version 2.0.1), which provided efficient handling of neural network training and backpropagation. Additionally, numerical computations and data handling were managed using NumPy (version 1.24.3), while data visualization and plotting were performed using Matplotlib (version 3.7.1). The simulation code leverages the random library for generating stochastic noise and initial conditions. Furthermore, the torch.optim module was employed to optimize the actor and critic networks using the Adam optimizer. To ensure the stability of the training process, the torch.nn module was used to construct the neural network architecture with ReLU activation functions.

Initially, high exploration noise is added to the actions taken by the actor to ensure sufficient exploration of the action space. Over time, this noise decays to encourage the agent to exploit learned behaviors. The reward is based on the SNR difference between the LU and the eavesdropper’s signal.

Results and discussion

This section assesses the performance of the proposed DDPG-based IRS-enabled secure communication system. To validate the efficacy of the proposed approach, comprehensive numerical analyses are conducted. The results demonstrate that our strategy achieves superior performance gains compared to existing methods like Deep-PDS and BCD-MM, even in challenging scenarios, by employing the unwanted jamming signals. In contrast to conventional IRS-based PLS techniques that only use IRS passive reflection, as well as the anti-eavesdropping technique that modifies the incoming signal into AN by using IRS backscatter, our proposed approach outshines in terms of performance. It can achieve superior performance, even in situations where the jammer is located near the user. The network layout described in Fig. 7 shows the location of BS, Jammer, eavesdropper and LU in the x-y coordinates. Specifically, their positions are given with respect to (0,0) locations and coordinates such as (0, 0) m for the BS, (35, 40) m for the eavesdropper, (60, 100) m for the IRS, (100, 40) m for the user, (130, 50) m for the jammer. We assume that all channels are Rayleigh fading channels and the noise power is δU2=δe2 = −90 dBm. We utilize the path loss model PL = PL0 − 30 log10(d/d0) dB, where d0 = 1 m serves as the reference distance and d is the distance between transmitter and receiver with the path loss PL0 = −30 dB.

Figure 7 The distances and locations of different entities in x-y plane.

We highlight the effectiveness of our proposed Deep-PLS method in enhancing security for LU against eavesdropping and jamming, by comparing it with the following schemes:

Deep PDS-PER enhanced the security of the communication system using deep reinforcement learning to optimize beamforming for multiple LU, ensuring secrecy rate and QoS (Yang et al., 2020). The Deep PDS-PER scheme is utilized to enhance the agent’s learning and exploration rate to satisfy the user’s minimum QoS requirements. In this scheme, multiple users and eavesdropper are considered in the system model but the impact of the jammer is not included, in the scenario, while the secrecy and data rate achieved was low due and the agent couldn’t show better convergence.

BCD-MM utilizes IRS to enhance wireless communication security, focusing on the challenges of dealing with eavesdropping and malicious jamming attacks in MIMO communication systems (Cao et al., 2023). This approach involves converting the incoming jammer’s interference signal into the intended signal using IRS backscatter. The BCD-MM algorithm successfully addressed the optimization problem and analyzed the influence of imperfect CSI on system performance. A conventional BCD-MM based optimization technique is used to convert non-convex optimization problems into a tractable form. DDPG algorithm can rapidly converge to an optimal solution as it learns directly from the environment. BCD-MM, on the other hand, requires iterative and computationally intensive updates for each subproblem.

Deep-PLS method is robust against dynamic environments and varying channel conditions, while Baseline schemes (Cao et al., 2023) method can fail under rapidly changing conditions. Due to adaptive learning and direct optimization, the secrecy rate achieved by Deep-PLS is statistically higher. The DRL-based DDPG algorithm is particularly suitable for this problem due to its ability to handle continuous action spaces, which is essential for optimizing beamforming vectors and IRS phase shifts. The dynamic nature of the environment, including time-varying channel conditions makes model-free reinforcement learning an ideal choice. The DDPG algorithm can adapt to these changes by learning directly from interactions with the environment, without requiring an explicit model of the system dynamics. The DRL framework allows for real-time decision-making, making it suitable for practical implementations where rapid adaptation to changing conditions is crucial.

Deep-PLS avoids local optima, scales better with IRS elements, and adapts to dynamic/imperfect CSI. DRL’s low-latency inference makes it suitable for real-time systems, unlike iterative BCD-MM. Deep-PLS achieves the (μ1 = 3.57 bits/s/Hz) while BCD-MM μ2 = 2.8 bits/s/Hz which proved Deep-PLS to be more significant.

The presented Fig. 8 compares the performance of three RL techniques i.e., Deep-PLS, Deep PDS-PER, and DQN evaluated based on their average reward across episodes. The results demonstrate that the Deep-PLS technique consistently achieves superior performance compared to both Deep PDS-PER and DQN, as reflected in the higher and more stable average rewards. This performance advantage is attributable to the use of the DDPG algorithm, which is particularly effective in environments requiring continuous action spaces. DDPG facilitates more efficient exploration and exploitation, leading to enhanced decision-making and security in IRS-assisted backscatter communication. The consistent outperformance of Deep-PLS underscores its suitability for this research, positioning it as the optimal choice over the alternative methods.

Figure 8 Comparison of different DRL techniques.

Figure 9 depicts episodes vs average reward, the x-axis shows the number of episodes, while the y-axis displays the average reward per episode. The graph displays the results for three learning rates: 0.01, 0.001, and 0.0001. It is evident that a learning rate of 0.01 yields the highest average reward per episode, with 0.001 and 0.0001 following closely behind. This implies that a higher learning rate enables the model to grasp concepts more quickly and attain a greater average reward at an earlier stage. Nevertheless, it is worth mentioning that a higher learning rate can result in instability and cause the model to diverge. The optimal learning rate can differ based on the particular task of reinforcement learning and the intricacy of the environment. It can be beneficial to try out various learning rates to discover the most effective one for your specific task.

Figure 9 Reward per episode vs different learning rates.

We initially examine the impact of the transmit power of BS on the secrecy rate. Figure 3 depicts a comparison of the secrecy rate achieved by three different techniques for improving wireless network security: the proposed Deep-PLS, the Deep-PDS, and the baseline numerical technique BCD-MM. The x-axis shows the maximum transmit power (PT) in decibel-milliwatt (dBm) units, and the y-axis represents the average secrecy rate (bits/s/Hz). The secrecy rate is the rate at which data is transmitted over a channel securely. The figure shows that the proposed Deep-PLS method has an average secrecy throughput that is higher than both the benchmark numerical methods, BCD-MM or Deep-PDS for all transmit power levels. It demonstrates the advantage of the DRL-based Deep-PLS method over the other techniques in improving wireless network security. While using 15 dBm as transmit power in the proposed Deep-PLS methodology, an average secrecy rate of around 0.7 bits/s/Hz is achieved. On the other hand, the average secrecy rate of over 3.57 bits/s/Hz is ensured using Deep-PLS at 40 dBm transmit power.

The Fig. 4 shows a comparison of the average secrecy rate achieved w.r.t to the jamming power. The X-axis shows jamming power (p/z) (dBm/Hz), and the y-axis shows the average secrecy rate (bits/s/Hz). The graph illustrates that the proposed Deep-PLS-based algorithm proves to be better in terms of average secrecy rate than not only the DL technique but also the numerical techniques for all the jamming power levels. The rate of successful concealment which is achieved by each technique shows very considerable difference depending on the power of jamming. It is to be noted that IRS installed in the system model between LU and Jammer, is backscattering the high-frequency jamming signal into a useful signal that maximizes the secrecy rate and the achievable data rate. This has been proven from the fig, When the jamming power is 15 dBm, the secrecy rate is at 2.49, 2.25, and 2 for DDPG, DQN, and numerical techniques, respectively. This means that the higher the jamming power, the more incident signals on IRS, and the better the backscattering to the LU, which ultimately enhances the secrecy rate and achievable data rate. However, this is to be expected, as factors might differ from one sharpening model to another in terms of the details and the settings being employed. However, the general tendency of the graph is very much probable that it may not be different, and the DRL-based DDPG technique can achieve a higher average secrecy rate than the other two techniques.

Figure 5 is the measure of the number of scattering elements (L) that can be viewed as mirrors or quantities 10 of elements used to steer the passage of radio waves. In the context of the graph, it is most probably the case that the reflective elements are helping to enhance the secrecy rate on a wireless network through the reflection of the signal by eavesdropper. The figure depicts that the Deep-PLS method exhibits a higher average secrecy rate for all the L-values concerning both other techniques, such as the Deep-PDS and the baseline numerical technique, BCD-MM. The x-axis goes in range from 40 to 100 including all ten numbers with ticks every tenth. Y-axis (Secrecy Rate (bps/Hz)) The x-axis connects to the point when it’s at 3 and finishes at 4, with labels on each 0.1 bits/s/Hz. Increasing the number of IRS elements enhances the IRS’s ability to beamforming. With this improved beamforming the desired secrecy rate can be achieved and also, a higher number of IRS elements backscatter the signals toward the LU, more effectively.

Figure 6 illustrates the achievable data rates RLU and Re vs the number of IRS elements L. As L increases, the rate RLU for the legitimate user improves significantly, indicating enhanced beamforming capability and more robust signal reflection towards the intended user. In contrast, the eavesdropper’s rate Re drops progressively, demonstrating the effectiveness of our DRL-based policy in suppressing unintended signal leakage. The error bars now included represent ±1 standard deviation from the mean data rates. These confidence intervals validate the consistency of our proposed Deep-PLS framework in achieving secure transmission.

The relatively narrow spread of the error bars, especially in RLU performance, indicates high reliability and low variance in the learned beamforming policy under varied channel realizations. Similarly, the stable suppression of Re across increasing IRS elements confirms that the optimization consistently degrades the eavesdropper’s rate. The performance separation between RLU and Re is therefore statistically robust, strengthening our argument for Deep-PLS effectiveness.

Conclusions and future works

In this article, we present a new strategy Deep-PLS to enhance backscatter in IRS To safeguard against the eavesdropping and jamming attempts in MISO communication systems with imperfect CSI. This approach converts the high-frequency jamming signals into useful signals using IRS backscatter technique. Our research focused on optimizing the beamforming of both the BS and the IRS in the context of time-varying channel conditions. Given the complex structure of the system, we have utilized the latest developments in machine learning to address the secure beamforming optimization problem using reinforcement learning techniques. Using this approach, we implemented a DRL-based DDPG technique to create the BS’s active beamforming and IRS backscatter coefficient matrix, intending to maximize the secrecy rate of wireless communication. The simulation results confirmed the significant ability of the IRS backscatter enhancing strategy to enhance system security and increase the secrecy rate. The proposed Deep-PLS is significantly superior to the traditional IRS-PLS technique, Deep-PDS, and BCD-MM for all settings.

While this work assumes ideal phase shifts and static users for tractability, real-world deployments often suffer from hardware imperfections and dynamic environments. In practice, RIS elements may experience phase noise, limited-resolution quantization, or phase drift due to hardware constraints. These impairments can reduce the effective beamforming gain and impact the policy learned by the DRL agent. Moreover, in highly mobile scenarios (e.g., UAV-assisted communication or fast-moving users), channel conditions may vary rapidly within the coherence time. This non-stationarity can degrade the performance of DRL algorithms that rely on temporal stability for policy convergence.

To address these concerns, future work could integrate online adaptation mechanisms or meta-learning strategies that allow the agent to quickly adapt to changes in environment dynamics. Incorporating robust DRL techniques or uncertainty-aware learning could also improve resilience to such impairments.

Appendix

Table of key variables and definitions.

Variable	Definition	Equation/Reference	
Gbr	Channel matrix between BS and IRS	Eqs. (1), (2)	
hbu	Channel vector between BS and legitimate user (LU)	Eq. (1)	
hbp	Channel vector between BS and eavesdropper	Eq. (2)	
gru	Channel vector between IRS and LU	Eq. (1)	
grp	Channel vector between IRS and eavesdropper	Eq. (2)	
Hjr	Channel matrix between jammer and IRS	Eq. (1)	
hju	Channel vector between jammer and LU	Eq. (1)	
h jp	Channel vector between jammer and eavesdropper	Eq. (2)	
w	Transmit beamforming vector at BS	Eqs. (1), (9)	
Φ	IRS reflection coefficient matrix (diagonal), with ωl (amplitude) and θl (phase shift)	Eqs. (1), (11)	
qj	Jamming signal vector	Eq. (1)	
yu, ye	Received signals at LU and eavesdropper, respectively	Eqs. (1), (2)	
nu, ne	Additive Gaussian noise at LU and eavesdropper (∼CN(0,δ2I)∼CN(0,δ2I)	Eqs. (1), (2)	
RLu, Re	Achievable data rates at LU and eavesdropper, respectively	Eqs. (3), (4)	
Rs	Secrecy rate (RLu − Re)	Eqs. (5), (6)	
h bu	Estimated channel error (modeled as Gaussian noise)	Eq. (7)	
ψ	Ratio of channel estimation error	Section 2.2	
βHbu	Path loss between BS and LU	Section 2.2	
Ωu, Ωe	Covariance matrices of noise for LU and eavesdropper	Eq. (8)	
Pt	Transmit power constraint at BS	Eq. (10)	

Supplemental Information

Supplemental Information 1 Code.

The authors are very grateful to the editors and all anonymous reviewers for their insightful comments.

Additional Information and Declarations

Competing Interests

The authors declare that they have no competing interests.

Author Contributions

Manzoor Ahmed conceived and designed the experiments, authored or reviewed drafts of the article, and approved the final draft.

Touseef Hussain conceived and designed the experiments, performed the experiments, analyzed the data, performed the computation work, prepared figures and/or tables, and approved the final draft.

Muhammad Shahwar performed the experiments, analyzed the data, prepared figures and/or tables, and approved the final draft.

Feroz Khan performed the experiments, prepared figures and/or tables, and approved the final draft.

Muhammad Sheraz conceived and designed the experiments, authored or reviewed drafts of the article, and approved the final draft.

Wali Ullah Khan performed the experiments, prepared figures and/or tables, and approved the final draft.

Teong Chee Chuah conceived and designed the experiments, authored or reviewed drafts of the article, and approved the final draft.

It Ee Lee conceived and designed the experiments, authored or reviewed drafts of the article, and approved the final draft.

Data Availability

The following information was supplied regarding data availability:

The code is available in the Supplemental File.

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
