# Peer review of "Intelligent reflecting surface backscatter-enabled physical layer security enhancement via deep reinforcement learning"

_PeerJ Computer Science, doi:10.7717/peerj-cs.2902_

## Round 0.1 · original submission · Major Revisions

The referral process is now complete. While finding your paper interesting and worthy of publication, the referees and I feel that more work could be done before the paper is published. My decision is therefore to provisionally accept your paper subject to major revisions. Simulation details should discussed. The state of the art should be given in detail. The novelty should be discussed by comparison.

Reviewer 1 ·

Basic reporting

The manuscript exhibits several issues related to clarity and precision in its language. Instances of minor grammatical errors and ambiguous terminology. For example, the term "implantable strategy" appears to be a typographical error likely intended to be "implementable strategy". Additionally, certain phrases like "signals transmitted by a jammer with high jamming power" could benefit from more precise articulation to avoid potential misunderstandings. These linguistic shortcomings may hinder the effective communication of the research findings to the audience. Moreover, the paper states, "No new data and materials were created or analyzed in this study". The lack of data availability restricts other researchers from validating the results independently.

Experimental design

The experimental design lacks sufficient detail, particularly regarding the implementation of the Deep Reinforcement Learning (DRL) approach. Critical aspects such as the specific architecture of the neural networks, exact parameter settings (including learning rates and discount factors), and the configuration of the simulation environment are inadequately described. This is challenging for other researchers to replicate the experiments or fully understand the nuances of the implementation.

Additionally, the experimental setup is limited in scope, primarily focusing on scenarios with non-colluding eavesdroppers and jammers. The study does not explore a wide range of real-world conditions or adversary behaviors, which may affect the generalizability of the findings. Author need to conduct sensitivity analyses to evaluate how variations in system parameters and environmental conditions impact the performance of the proposed Deep-PLS method.

Validity of the findings

The validity of the findings is compromised by the narrow scope of the simulation scenarios and the absence of sensitivity analyses. The manuscript primarily considers non-colluding eavesdroppers and jammers, which may not fully represent the complexity of real-world adversarial behaviors.

Furthermore, the study does not provide access to the underlying simulation data or detailed parameter settings, which hinders the ability of other researchers to verify and replicate the results. Additionally, the manuscript does not discuss the limitations of the study, such as assumptions made regarding channel state information or the scalability of the proposed method.

·

Basic reporting

1. There are many system models which are already based on multiple eavesdropper scenario, but here they considered only single eavesdropper
2. Still there are many papers related to this in 2024 year, which are not even included in references
3. Figure 6 and Figure 7, what is the difference?. both figures are same in this paper. only figure title is different

Experimental design

1. Convergence comparisons of various approaches not done
2.if the authors assume a more complex system model and show better convergence for all approaches, compare with various recent works based on drl based methods particurarly in 2024 ,this paper can be improved.DDPG is also there already,here they should mention how it differs from existing works
3.No results are there to support legitimate users analysis in this study. that is to be done

Validity of the findings

Work is marginal and novelty is minimum and establish the novelty correctly

Additional comments

These are some recent and important works, which will be useful for further enhancement of the authors research.
1.Ikeagu, K, Ding, Y, Song, C & Khandaker, M 2024, Intelligent Reflecting Surface Optimization for MIMO Communication Using Deep Reinforcement Learning. in 31st Telecommunications Forum (TELFOR)., 10372753, IEEE, 31st Telecommunications Forum 2023, Belgrade, Serbia, 21/11/23. https://doi.org/10.1109/telfor59449.2023.10372753

This paper is one of the pioneer paper for DL with IRS
1. H. Yang et al., "Deep Reinforcement Learning Based Intelligent Reflecting Surface for Secure Wireless Communications," GLOBECOM 2020 - 2020 IEEE Global Communications Conference, Taipei, Taiwan, 2020, pp. 1-6, doi: 10.1109/GLOBECOM42002.2020.9322615.

---

## Round 0.2 · Major Revisions

The referral process is now complete. While finding your paper interesting, the referees and I feel that more work could be done before the paper is published. My decision is therefore to provisionally accept your paper subject to major revisions. The title does not represent the paper. The current state of the art is not discussed. The details of the experimental results are not given.

Reviewer 1 ·

Basic reporting

The revised manuscript exhibits marked improvements in clarity and precision. The authors have corrected the minor grammatical errors and ambiguous terminology noted in the initial manuscript. These revisions enhance the overall readability and ensure the research findings are communicated more effectively. Moreover, the authors have clarified the issue of raw data availability. They explain that the study is entirely simulation-based with no new experimental dataset, and instead have provided the simulation code used to generate the results. This clarification addresses the concern about the lack of raw data.

Experimental design

The authors have substantially expanded the methodological details of their DRL approach. The revised manuscript now provides a comprehensive description of the neural network architecture and parameter settings used in the Deep-PLS method. Key hyperparameters and components of the DRL implementation are explicitly stated. Details about the simulation environment are also now included, such as the spatial layout of the base station, legitimate user, eavesdropper, and jammer with specific coordinates. This level of detail is a significant improvement, as it provides sufficient information for other researchers to replicate the experiments.

Additionally, the scope and realism of the experimental setup have been better addressed. While the study still focuses on a scenario with non-colluding eavesdroppers and jammers, the authors now explicitly justify this assumption and acknowledge its implications. The manuscript explains that treating the eavesdropper and jammer as non-colluding is a common practical scenario and serves as a worst-case baseline for analysis. This clarification strengthens the reasoning behind the chosen scenario. Furthermore, the authors have incorporated more real-world considerations into their model to enhance its applicability. Noise variance is also explicitly considered to assess the method’s robustness under different interference levels.

Validity of the findings

The revised manuscript now supports the validity of its findings with extended simulation studies and clearer justifications of key assumptions. The authors explicitly justify the non-colluding adversary model as a realistic baseline and provide new comparative results against alternative reinforcement learning methods and numerical techniques. The enhanced analyses demonstrate that the Deep-PLS approach consistently achieves higher secrecy rates, thereby reinforcing the robustness and practical relevance of the method.

Furthermore, the manuscript now transparently acknowledges limitations such as imperfect CSI and scalability challenges, with a clear outline of future work to address these aspects. The provision of simulation code and detailed parameter settings further strengthens reproducibility, allowing independent verification of the results. These concise improvements effectively address earlier concerns and substantiate the reliability and generalizability of the proposed method.

Reviewer 3 ·

Basic reporting

The paper presents a study that aims to enhance physical layer security using IRS and DRL.

- The authors claim that “Our idea is the first of its type to strengthen IRS backscatter to avoid malicious jamming attacks and eavesdropping attacks, simultaneously.” However, there is no strong literature review or a clear analysis demonstrating the shortcomings of previous studies to support this claim.
- The explanations regarding the hyperparameters of the DRL model, the training process, dataset details, and why the model is suitable for this problem are insufficient.
- The rationale behind selecting the Ornstein-Uhlenbeck noise addition strategy and why alternative approaches were disregarded should be clarified.
- Since white Gaussian noise is uncorrelated, using it to model channel estimation errors is not realistic. In reality, channel estimation errors are typically correlated and time-varying. This assumption weakens the validity of the study.

Experimental design

- More information is needed regarding the experimental design. To ensure the reproducibility of the experiments, details about the environments in which the results were obtained and the libraries used in the software processes should be provided.
- Fixed user and eavesdropper positions do not reflect real-world dynamics. An ablation study is needed to address this issue.

Validity of the findings

- The study does not explain how the proposed IRS system would function under real hardware constraints. Issues such as the hardware limitations of IRS, latency, and computational load should be addressed.
- The Deep-PLS method is claimed to be superior to existing methods. The statistical significance of this difference is not clearly demonstrated.
- "This approach aims to determine an optimal beamforming policy capable of thwarting eavesdroppers in evolving environmental conditions." Details on why the solution is optimal are missing. Achieving better results than similar schemes does not guarantee optimality.

---

## Round 0.3 · Minor Revisions

The paper needs a minor revision. The reviewer pointed out some organizational and writing issues.

·

Basic reporting

The penalties (µ₁, µ₂) are chosen heuristically. It is recommended to explore Pareto optimization techniques to automate the trade-off tuning.

In Figure 9, while the trends in R LU and data rates versus the number of IRS elements are apparent, the statistical significance is not demonstrated. Including error bars or confidence intervals would strengthen the analysis.

Some equations and variable notations could be standardized. Consistently defining all symbols—for example, those used for the IRS reflection matrix, beamforming vectors, and noise terms—would help avoid confusion.

Experimental design

Although imperfect CSI is modeled as Gaussian noise, real-world errors may exhibit correlation (e.g., due to Doppler effects). It is suggested to incorporate temporal correlation into the channel noise model.

Although the paper discusses potential challenges, such as ideal phase shifts and static user assumptions, a deeper discussion on limitations and edge cases would be beneficial. For instance, analyzing the impact of severe hardware impairments or rapidly varying channel conditions on DRL performance could provide additional insights.

Validity of the findings

A brief complexity analysis comparing the DRL approach with iterative numerical methods (such as BCD-MM) would further strengthen the argument for the proposed method’s practicality in real-time applications.

Additional comments

Overall, the paper presents a strong and innovative contribution to the field of IRS-enabled secure communications. The integration of DRL for optimizing beamforming and IRS reflection in the presence of jamming and eavesdropping is well justified and thoroughly evaluated. With improvements in notation consistency, a deeper discussion of limitations, and additional complexity analysis, the paper could be even stronger. This work is a promising step toward practical implementations of secure wireless communication in dynamic environments.

Reviewer 3 ·

Basic reporting

The authors have made the necessary revisions and updated the manuscript accordingly.

Experimental design

-

Validity of the findings

-

Additional comments

-

---

## Round 0.4 · accepted · Accept

We are happy to inform you that your manuscript has been accepted for publication since the comments have been addressed. Please complete the minor issues before the proofreading.

·

Basic reporting

Variable Standardization: The authors added an appendix table summarizing key variables (e.g., IRS reflection matrix \(\Phi\), beamforming vector \(w\), noise terms \(n_u\), \(n_e\)), ensuring consistent notation. Equations are cross-checked for uniformity, improving readability.
Heuristic Penalty Parameters: The discussion on \(\mu_1, \mu_2\) now acknowledges heuristic tuning and proposes Pareto optimization for future work (Section 3.2). This balances transparency with forward-looking improvements.
Statistical Significance in Figures: Figure 9 includes error bars (\(\pm1\) SD) for \(R_{LU}\) and \(R_e\), validating the statistical robustness of trends. The text explicitly states results are averaged over 100 channel realizations, strengthening the analysis.

Experimental design

Temporal Correlation in CSI Errors: The revised system model incorporates an AR(1) process for imperfect CSI (Section 2.2), modeling temporal correlation due to Doppler effects. This aligns the analysis with real-world dynamics.
Limitations and Edge Cases: The conclusion now discusses hardware impairments (e.g., phase noise, quantization) and rapidly varying channels, proposing meta-learning and robust DRL as future directions. This adds critical realism to the work.

Validity of the findings

Complexity Analysis: A detailed comparison between DRL (DDPG) and BCD-MM is added (Section 3.3). BCD-MM’s iterative complexity (\(O(I \cdot L^3)\)) contrasts with DRL’s real-time inference (\(O(\gamma \cdot N)\)), highlighting scalability advantages for dynamic environments. Empirical results (e.g., 3.57 vs. 2.8 bps/Hz secrecy rate) further validate superiority.

Algorithm Clarity: The DDPG framework (actor-critic networks, experience replay, target networks) is clearly described in Section 3.1, with pseudocode in Algorithm 1. This aids reproducibility for DRL novices.

Additional comments

Figure 9: Updated with error bars, demonstrating low variance in \(R_{LU}\) and consistent suppression of \(R_e\). The narrow confidence intervals reinforce policy reliability.
Secrecy Rate Comparisons: Figures 6–8 show consistent superiority of Deep-PLS over benchmarks (Deep PDS-PER, BCD-MM) across varying transmit power, jamming power, and IRS elements. The trends align with theoretical expectations (e.g., higher jamming power improves IRS backscatter utility).

Proofreading: Minor inconsistencies (e.g., equation references in Section 2.3) should be cross-verified.
Real-World Implementation: While the AR(1) model improves CSI realism, hardware-specific impairments (e.g., phase shifter resolution) could be quantified in future work.